# Efficient Heat Transfer Augmentation in Channels with Semicircle Ribs and Hybrid Al_2_O_3_-Cu/Water Nanofluids

**DOI:** 10.3390/nano12152720

**Published:** 2022-08-07

**Authors:** Hussein Togun, Raad Z. Homod, Zaher Mundher Yaseen, Azher M. Abed, Jameel M. Dhabab, Raed Khalid Ibrahem, Sami Dhahbi, Mohammad Mehdi Rashidi, Goodarz Ahmadi, Wahiba Yaïci, Jasim M. Mahdi

**Affiliations:** 1Department of Biomedical Engineering, University of Thi-Qar, Nassiriya 64001, Iraq; 2College of Engineering, University of Warith Al-Anbiyaa, Karbala 56001, Iraq; 3Department of Oil and Gas Engineering, Basrah University for Oil and Gas, Basrah 61004, Iraq; 4Department of Earth Sciences and Environment, Faculty of Science and Technology, Universiti Kebangsaan Malaysia, Bangi 43600, Selangor, Malaysia; 5Adjunct Research Fellow, USQ’s Advanced Data Analytics Research Group, School of Mathematics Physics and Computing, University of Southern Queensland, Toowoomba, QLD 4350, Australia; 6New era and Development in Civil Engineering Research Group, Scientific Research Center, Al-Ayen University, Thi-Qar 64001, Iraq; 7Department of Air Conditioning and Refrigeration, Al-Mustaqbal University College, Babylon 51001, Iraq; 8Alnukhba University College, Baghdad, Iraq; 9Department of Medical Instrumentation Engineering, Al-Farahidi University, Baghdad 10015, Iraq; 10Department of Computer Science, College of Science and Art at Mahayil, King Khalid University, Aseer 62529, Saudi Arabia; 11Institute of Fundamental and Frontier Sciences, University of Electronic Science and Technology of China, Chengdu 610054, China; 12Faculty of Science, University of Johannesburg, P.O. Box 524, Auckland Park 2006, South Africa; 13Department of Mechanical and Aerospace Engineering, Clarkson University, Potsdam, NY 13699-5725, USA; 14CanmetENERGY Research Centre, Natural Resources Canada, 1 Haanel Drive, Ottawa, ON K1A 1M1, Canada; 15Department of Energy Engineering, University of Baghdad, Baghdad 10071, Iraq

**Keywords:** hybrid nanofluids, thermal performance, ribs channel, turbulent flow, separated flow

## Abstract

Global technological advancements drive daily energy consumption, generating additional carbon-induced climate challenges. Modifying process parameters, optimizing design, and employing high-performance working fluids are among the techniques offered by researchers for improving the thermal efficiency of heating and cooling systems. This study investigates the heat transfer enhancement of hybrid “Al_2_O_3_-Cu/water” nanofluids flowing in a two-dimensional channel with semicircle ribs. The novelty of this research is in employing semicircle ribs combined with hybrid nanofluids in turbulent flow regimes. A computer modeling approach using a finite volume approach with k-ω shear stress transport turbulence model was used in these simulations. Six cases with varying rib step heights and pitch gaps, with Re numbers ranging from 10,000 to 25,000, were explored for various volume concentrations of hybrid nanofluids Al_2_O_3_-Cu/water (0.33%, 0.75%, 1%, and 2%). The simulation results showed that the presence of ribs enhanced the heat transfer in the passage. The Nusselt number increased when the solid volume fraction of “Al_2_O_3_-Cu/water” hybrid nanofluids and the Re number increased. The Nu number reached its maximum value at a 2 percent solid volume fraction for a Reynolds number of 25,000. The local pressure coefficient also improved as the Re number and volume concentration of “Al_2_O_3_-Cu/water” hybrid nanofluids increased. The creation of recirculation zones after and before each rib was observed in the velocity and temperature contours. A higher number of ribs was also shown to result in a larger number of recirculation zones, increasing the thermal performance.

## 1. Introduction

Due to global technological advances, energy demand is rising daily, despite the associated carbon-induced climate challenges. The researchers have introduced a range of practical methods to increase the thermal efficiency of heating and cooling systems, including adjusting process parameters, design optimization, and the use of high-performance working fluids. In recent years, there have been significant advances in using nanofluids and hybrid nanofluids with thermal conductivities that are significantly higher than normal fluids. Smulsky et al. [1] investigated fluid flow and heat transfer in a pipe with 50- to 90-degree rib angles. They discovered that rib height and angle markedly affect thermal and flow characteristics in the pipe, with the greatest heat transfer performance achieved at rib angles of 50 degrees. Caliskan and Baskaya [2] studied heat transfer in a circular jet array with Re values ranging from 2000 to 10,000 using “V-shaped” ribs and convergent- and divergent-formed ribs. The gain in heat transfer for V-SR rose from 4% to 26% compared to other cases. Nine et al. [3] investigated turbulent duct flows and the accompanying friction factors in a duct with semicircle ribs. They found the numerical predictions are correlated well with the experimental data regarding the influence of rib structure on heat transfer improvement. Togun et al. [4] examined turbulent fluid flow and heat transfer in a pipe with a half-circle rib using the “SST k-turbulence model.” They discovered that increasing step height and rib pitch ratio improves thermal efficiency. Riyadh et al. [5] used computational and experimental methods to investigate the effect of the height of semicircle ribs in pipes on thermal performance with turbulent two-phase flows. The authors discovered that ribbed channels have a higher heat transfer rate than smooth channels.

Nanofluids with fabricated geometries are increasingly employed to improve heat transfer in various applications [6,7,8,9,10,11,12,13,14,15,16,17,18,19,20]. Dadheech et al. [6] studied natural convection in the attendance of an angled magnetic field and compared the thermal behavior of MoS_2_/C2H_6_O_2_ and SiO_2_-MoS_2_/C_2_H_6_O_2_ nanofluids. They solved the governing equations using suitable similarity transformations and the fourth-order Runge–Kutta method to evaluate the temperature and velocity profiles. They concluded that increasing the convection parameter enhances the velocity profile while negatively impacting the temperature profile for both nanofluids. Additionally, they found enhanced velocity and temperature profiles for both nanofluids when the volume fraction is raised. Heat transport analysis of water-based Al_2_O_3_ nanofluids and water-based CuO nanofluids on an accelerated radiative Riga plate surface was conducted by Kanayo et al. [7]. The governing equations of the flow were solved using the Laplace transform approach. The effects of several variables on Nu, C_f_, temp., and velocity distribution are investigated, and the results were presented in tabular and graphical forms.

Oronzio et al. [21] examined nanofluid flow and heat transmission in ribbed channels numerically. They reported that the use of nanofluid in a pipe with ribs resulted in further increases in heat transfer rates. Mohammed et al. [22] studied the hydraulic and thermal features of turbulent nanofluids’ flow in a pipe with rib–grooves. In their study, they used three base fluids, “water, glycerin, and engine oil”, as well as “Al_2_O_3_, CuO, SiO_2_, and ZnO” nanoparticles with concentrations ranging from 1% to 4%. They reported the influences of aspect ratio, Re, and volume concentration on Nu increases and improved thermal efficiency of nanofluids. Turbulent nanofluid thermo-fluidic characteristics in a pipe with half-circle ribs were investigated by Togun [23]. The effects of step height, the solid volume concentration of nanofluids, and Reynolds on heat transfer rate improvement were investigated using the SST k-ω turbulence model. The highest heat transfer coefficient was found for a step height of 5 mm, 4 percent Al_2_O_3_ nanofluids, and a Reynolds number of 25,000. Khadija et al. [24] considered the influence of gaps between ribs in a micro-channel on thermal efficiency for nanofluids’ flow. They reported that increases in the Reynolds number and solid volume fraction of nanofluids, as well as a reduction in the gaps between the ribs, led to improved thermal efficiency. Mohammad et al. [25] investigated turbulent heat transfer and nanofluid flow in a rectangular ribbed channel with ribs at various angles. They discovered that a 60-degree attack angle resulted in a greater increase in heat transfer augmentation. Raheem et al. [26] conducted a mathematical analysis on turbulent SiO_2_ nanofluid flow and heat transfer in a “semicircle-corrugated” pipe. Nusselt number was shown to rise with increases in rib height, solid volume%, and Re. Pushpa et al. [27] developed a computer model to investigate the buoyancy-driven flow and heat transport enhancement of Cu–H_2_O nanofluids in an upright annular cavity with a thin baffle. The governing equations are solved using a finite difference-based numerical approach. Their results were presented in terms of isotherms, streamlines, and Nu numbers over a range of baffle placements and lengths, Rayleigh numbers, and the nanofluid solid volume fractions. They reported that the average Nu number increased when the Cu nanoparticle was added to the base liquid. The liquid flow and heat transfer were successfully regulated by selecting the proper baffle placement and length.

Hybrid nanofluids are a novel type of nanofluid generated by mixing “two or more” types of nanoparticles in a mixture or compound form and are currently being utilized to enhance thermal efficiency. Bahiraei [28] provided a computational simulation of the energy efficiency and hydrothermal properties of MWCNT-Fe_3_O_4_/water hybrid nanofluid flow through a triple-tube with ribs. Their findings revealed that growing the rib height and volume fractions of hybrid nanofluids significantly impacts thermal performance. Rasul et al. [29] applied the LBM to study the effect of MWCNT-Fe_3_O_4_/ hybrid nanofluid and ribs on heat transfer rate enhancement. In all cases, the highest heat transfer coefficient was about 16.5%. Numerous investigations have recently been adopted on the effects of hybrid nanofluids and ribs in microchannels [30,31,32,33,34,35]. Additional applications of nanofluids in engineering problems were reported in [36,37,38,39,40].

The presented literature survey shows that few researchers have studied the hybrid nanofluid flows in ribbed channels. Additionally, the effectiveness of semicircle ribs has not been investigated previously. Therefore, the present study investigated the effects of volume concentrations of hybrid Al_2_O_3_-Cu/water nanofluids, semicircle ribs with different step heights and pitch ratios, and Re number on heat transfer enhancement. The novelty of this study is in using semicircle ribs combined with hybrid nanofluids in the turbulent flow regime. The study showed that increasing the size and number of ribs resulted in significant improvements in Nusselt number. Therefore, it is conjectured that heat transfer enhancement is due to increases in the number and size of recirculation regions, which significantly impact the overall thermal efficiency.

## 2. Physical Model and Mathematical Formulation

The channel considered by Togun [23] is adopted to create duct geometry, as presented in Figure 1. The total length of the channel is 1000 mm and the height of H = 40 mm, with semicircle ribs on the wall at 2.5 mm and 5 mm step heights with spacing ranging from 5 to 10 mm. In this analysis, six cases were applied with pitch ratios ranging from 10 to 40 for 2.5 mm rib height and 5 to 20 for 5 mm rib height; for more details, see Table 1. With water as the base fluid, four hybrid Al_2_O_3_-Cu/water nanofluids with solid volume fractions of 0.33, 0.75, 1, and 2% were considered. For a constant wall temperature of 320 K, the flow Re numbers varied from 10,000 to 25,000. The heat transfer characteristics of hybrid nanofluids were defined using the single-phase model.

In this study, the hybrid nanofluids flow across the heated test section and absorb heat from the channel’s hot top and bottom surface. In this investigation, the following assumptions were made:A two-dimensional computational domain is assumed to allow the flow domain to be treated as two parallel plates with ribs.The temperature dependence of thermal conductivity is negligible.The hybrid nanofluid steady-state flow is in the turbulent flow domain.A single-phase model was used to simulate hybrid nanofluids.

The generalizations to three-dimensional simulations and the inclusion of the effect of temperature-dependence material properties are left for future studies.

The single-phase model’s mass conservation, momentum balance, and energy conservation equations are stated as follows [21]:(1)$$\rho \frac{\partial }{\partial x_{i}}\left(u_{i}\right)=0$$
(2)$$\rho \frac{\partial }{\partial x_{j}}\left(u_{i}u_{j}\right)=-\frac{\partial P}{\partial x_{i}}+\frac{\partial }{\partial x_{i}}\left[\mu \left(\frac{\partial u_{i}}{\partial x_{j}}+\frac{\partial u_{j}}{\partial x_{i}}\right)\right]+\frac{\partial }{\partial x_{j}}\left(-\rho \bar{u_{i}^{\prime }u_{j}^{\prime }}\right)$$
(3)$$\frac{\partial }{\partial x_{i}}\left[u_{i}\left(\rho E\right)+P\right]=\frac{\partial }{\partial x_{j}}\left[\left(\lambda +\frac{c_{p}\mu_{t}}{{\mathrm{Pr}}_{t}}\right)\frac{\partial T}{\partial x_{j}}+u_{i}{\left(\tau_{\mathrm{ij}}\right)}_{\mathrm{eff}}\right]$$
here $u_{i}$ is velocity vector in i direction, m/s, $\rho ,\mu ,\lambda ,{\;\mathrm{and}\;c}_{p}$ represent the density, the dynamic viscosity, thermal conductivity, and specific heat at constant pressure, respectively, for the nanofluid. In Equation (2), $-\rho \bar{u_{i}^{\prime }u_{j}^{\prime }}$ denote the Reynolds stress tensor. In Equation (3), $E=T+\left(u^{2}/2\right)$, and ${\left(\tau_{\mathrm{ij}}\right)}_{\mathrm{eff}}$ describes the deviatoric stress tensor given by,
(4)$${\left(\tau_{\mathrm{ij}}\right)}_{\mathrm{eff}}=\mu_{\mathrm{eff}\;}\left(\frac{\partial u_{j}}{\partial x_{i}}+\frac{\partial u_{i}}{\partial x_{j}}\right)$$

Menter’s [41] SST k-ω-turbulence model, which was utilized in [42,43,44], can be written as:(5)$$\rho \left(\frac{\partial }{\partial x_{i}}\left({\mathrm{ku}}_{i}\right)\right)=\frac{\partial }{\partial x_{j}}\left(\left(\mu +\frac{\mu_{t}}{\sigma_{k}}\right)\frac{\partial k}{\partial x_{j}}\right)+G_{k}-Y_{k}$$
(6)$$\rho \left(\frac{\partial }{\partial x_{i}}\left({\omega u}_{i}\right)\right)=\frac{\partial }{\partial x_{j}}\left(\left(\mu +\frac{\mu_{t}}{\sigma_{\omega }}\right)\frac{\partial \omega }{\partial x_{j}}\right)+G_{\omega }-Y_{\omega }+D_{\omega }$$
where G_k_ denote stands for the generation of turbulent kinetic energy and D_ω_ signifies the cross diffusional terms, G_ω_ is the production of ω, and Y_k_ and Y_ω_ denote the dissipation of k and ω, while $\mu_{t}$ is given by,
(7)$$\mu_{t}=\propto^{*}\frac{\rho k}{\omega }$$
and $\propto^{*}$ is calculated as:(8)∝*=∝∞*(∝0*+Ret/Rk)(1+Ret/Rk)
here, Ret=ρk/μω, $\propto_{0}^{*}=\beta_{i}/3$ and
(9)$$\beta_{i}=F_{1}\beta_{i,1}+\left(1-F_{1}\right)\beta_{i,2}$$

The constants in the k and ω equations are given by [45]:(10)$$\sigma_{k}=\frac{1}{F_{1}/\sigma_{k,1}+\left((1-F_{1})/\sigma_{k,2}\right)}$$
(11)$$\sigma_{\omega }=\frac{1}{F_{1}/\sigma_{\omega ,1}+\left((1-F_{1})/\sigma_{\omega ,2}\right)}$$
here σ represents the turbulent Prandtl number of k and ω

The blending function (F1) is given as,
(12)$$F_{1}=\tan h\left(\Phi_{1}^{4}\right)$$
where
(13)$$\Phi_{1}=\min\left[\max\left(\frac{\sqrt{k}}{0.09\omega y}\frac{500\mu }{{\rho y}^{2}\omega }\right),\frac{4\rho k}{\sigma_{\omega ,2}D_{\omega }^{+}y^{2}}\right]$$
and
(14)$$D_{\omega }^{+}=\max\left[2\rho \frac{1}{\sigma_{\omega ,2}}\frac{1}{\omega }\frac{\partial k}{\partial x_{i}}\frac{\partial \omega }{\partial x_{j}},10^{-10}\right]$$
here $\Phi_{1}$ is pressure–strain term and $D_{\omega }^{+}$ is the part of the cross-diffusion term and is positive.

In Equation (5), G_k_ stands for the generation of turbulent kinetic energy due to the mean velocity gradients, while G_ω_ means the production of ω and D_ω_ signifies the cross diffusional terms.
(15)$$D_{\omega }=2\left(1-F_{1}\right){\rho \sigma }_{\omega ,2}\frac{1}{\omega }\frac{\partial k}{\partial x_{j}}\frac{\partial \omega }{\partial x_{i}}$$
Y_k_ and Y_ω_ display the dissipation of k and ω, and are given as,
(16)$$Y_{k}={\;\rho \beta }^{*}k\omega$$
(17)$$Y_{\omega }={\;\rho \beta }_{i}\omega^{2}$$
G_k_ can be found by:(18)$$G_{k}=\tau_{t,\mathrm{ij}}\left(\partial u_{i}/\partial u_{j}\right)$$
where:(19)$$\tau_{t,\mathrm{ij}}=\mu_{t}\left(\frac{\partial u_{i}}{\partial x_{j}}+\frac{\partial u_{j}}{\partial x_{i}}\right)-\left(\frac{2}{3}{\rho k\delta }_{\mathrm{ij}}\right)$$
$G_{\omega }$ is also a function of $G_{k}\;$:(20)$$G_{\omega }=\left(\rho \alpha /\mu_{t}\right)G_{k}$$
here ∝ is given as:(21)∝=(α∞α*)(∝0*+Ret/Rk)(1+Ret/Rk)
where $\propto_{\infty }$ is determined by:(22)$$\propto_{\infty }={\;F}_{1}\propto_{\infty ,1}+\left(1-F_{1}\right)\propto_{\infty ,2}$$
where
(23)$$\propto_{\infty ,1}=\frac{\beta_{i,1}}{\beta_{\infty }^{*}}-\frac{k^{2}}{\sigma_{\omega ,1}\sqrt{\beta_{\infty }^{*}}}$$
(24)$$\propto_{\infty ,2}=\frac{\beta_{i,2}}{\beta_{\infty }^{*}}-\frac{k^{2}}{\sigma_{\omega ,2}\sqrt{\beta_{\infty }^{*}}}$$

Table 2 shows the coefficients of the SST K- ω model.

## 3. Characteristics of Hybrid Nanofluids

The hybrid nanofluids used in this study are made of “Al_2_O_3_-Cu nanoparticles” suspended in water as a base liquid. We have selected the hybrid nanofluids using Al_2_O_3_ and Cu nanomaterials in our modeling study because these hybrid nanofluids are known to generate significant enhancement in heat transfer. Here, a single-phase, incompressible, Newtonian fluid model is used. The thermophysical characteristics of water, alumina, and copper are shown in Table 3 [46]. Table 4 [47] shows the thermal conductivity and dynamic viscosity of hybrid nanofluids as a function of solid volume percentage. The density and specific heat capacity of Al_2_O_3_-Cu/water hybrid nanofluids are calculated using the mixing model [41]. That is,
(25)$$\rho_{\mathrm{nf}}=\varphi_{\mathrm{cu}}\rho_{\mathrm{cu}}+\varphi_{{\mathrm{Al}}_{2}O_{3}}\rho_{{\mathrm{Al}}_{2}O_{3}}+\left(1-\varphi \right)\rho_{f}$$
(26)$${\left({\rho c}_{p}\right)}_{\mathrm{nf}}=\left(1-\varphi \right){\left({\rho c}_{p}\right)}_{f}+\varphi_{\mathrm{cu}}{\left({\rho c}_{p}\right)}_{\mathrm{Cu}}+\varphi_{{\mathrm{Al}}_{2}O_{3}}{\left({\rho c}_{p}\right)}_{{\mathrm{Al}}_{2}O_{3}}$$

The combined solid volume fraction of alumina and copper nanoparticles is given as,
(27)$$\varphi ={\;\varphi }_{\mathrm{Cu}}+\varphi_{{\mathrm{Al}}_{2}O_{3}}$$

## 4. Numerical Method and Validation

A finite volume-based computational fluid dynamics program, namely, ANSYS-FLUENT code, was used for simulations in this study. The standard SST k-ω-turbulence model, one of the most popular turbulence models, is utilized because it provides a better estimate of flow separation and flow behavior under adverse pressure gradients. The SIMPLE technique with pressure–velocity coupling was used to solve the energy and momentum equations with a second-order upwind scheme. The velocity component and energy residuals were set to 10^−8^ and 10^−11^, respectively, as a strategy for generating high-precision data. To obtain grid independence solutions, three different mesh sizes of 40,480, 92,400, and 165,760 nodes were used. The simulated average heat transfer coefficients for pure water at “h/H = 0.12, p/w = 10, and Re = 10,000” were evaluated and compared in Table 5. The average heat transfer coefficient for the second and third meshes is relatively modest, as seen in this table; as a result, the second mesh was chosen and utilized for the subsequent simulations.

The numerical results for airflow in a pipe with semicircle ribs studied by Togun et al. [4] are employed to validate the current computational model. It should be noted that the experimental data for turbulent nanofluid flow and heat transfer in a channel with semicircle ribs have yet to be published in the literature. The present model simulated the water flow for the same geometry, which was considered by Togun et al. [4] at Re = 25,000. Figure 2 shows that the spatial variations of the current model predictions for the local heat transfer coefficient for water flow are similar to those of Togun et al. [4] for air flows.

## 5. Results and Discussion

### 5.1. Impact of Reynolds Number

The impact of Reynolds number on variations of Nusselt number at a step height of 5 mm for pitch ratios (P/W) of 10 and 5 for the solid volume fraction of Al_2_O_3_-Cu/water hybrid nanofluids of 2% was shown in Figure 3A,B. In both cases, the Nu number is augmented as the Re number rises, indicating an increase in turbulence and recirculation flows. In addition, the formation of peaks between every two ribs demonstrates the same patterns, showing an increase in heat transfer. The largest Nusselt number is noticed for Re = 25,000, which is the highest Reynolds number studied.

### 5.2. Impact of Pitch Ratio

Figure 3C,D display the influence of pitch ratio on Nu number variations for Reynolds numbers of 25,000 and volume concentrations of Al_2_O_3_-Cu/water hybrid nanofluids of 2%, respectively, for step height ratios (h/H) of 0.06 and 0.12. The number of peaks in the profile of the local Nu increased as the pitch ratio decreased for both step height ratios (h/H) of 0.06 and 0.12, indicating an increase in heat transfer rate due to an increase in the number of ribs generating recirculation.

### 5.3. Impact of Using Hybrid Nanofluids

Re = 25,000, Figure 4A,B show the effects of volume concentration of Al_2_O_3_-Cu/water hybrid nanofluids on the local Nusselt number at pitch ratios of 5 (h/H = 0.12) and 10 (h/H = 0.06), respectively. The local Nu number increases with increasing solid volume concentrations of Al_2_O_3_-Cu/water hybrid nanofluids for both cases, as shown in these figures. The heat transfer is also improved by increasing the concentration of hybrid nanoparticles for both cases, with the maximum Nusselt number detected for 2% solid volume fraction of Al_2_O_3_-Cu/water hybrid nanofluids, which is the highest concentration for each case. For more clarity, both figures included zooms near the peak values to show the effects of “Al_2_O_3_-Cu/water” hybrid nanofluid volume concentration on the local Nusselt number.

### 5.4. Impact of Step Height

In Figure 5, all cases of current investigations are potted for different step heights of ribs and pitch space at a Re number of 25,000 and 2% solid volume fraction of hybrid nanofluids. In general, the results showed that for all pitch spaces, the local Nu number with a step height of 5 mm (h/H = 0.12) is higher than with a step height of 2.5 mm (h/H = 0.06) due to a rise in rib height. Increases in the number and size of ribs resulted in the greatest improvement in Nu number due to increases in the number and size of the recirculation zones, which significantly impacts thermal efficiency.

### 5.5. Pressure Coefficient

Figure 6A,B display the variation of pressure coefficient along the channel for changed Re numbers and solid volume fractions of hybrid nanofluids. Here, a step height ratio (h/H) of 0.12 and a pitch ratio (P/W) of 10 are assumed. These figures show that the local pressure coefficient generally decreases along the channel. In addition, the effects of variation of the Re number and solid volume fraction of hybrid nanofluids on the local pressure coefficient are negligibly small.

## 6. Conclusions

The enhancement of heat transfer through channels with semicircle ribs using hybrid Al_2_O_3_-Cu/water nanofluids was studied. Six cases with different rib step heights and pitch spacings were used in the analysis in addition to four different volume concentrations of hybrid nanofluids Al_2_O_3_-Cu/water (0.33, 0.75, 1, and 2%), with Re numbers ranging from 10,000 to 25,000. The study revealed that increasing the number and size of ribs resulted in the largest improvement in Nusselt number. This was due to an increase in the number and size of recirculation zones, which substantially influenced thermal efficiency. In addition, the Nusselt number increased as the solid volume concentrations of Al_2_O_3_-Cu/water hybrid nanofluids and the Reynolds number increased. The greatest Nu number occurred at 2% solid concentration hybrid nanofluids and for a Reynolds number of 25,000, the highest concentration and Reynolds number employed. As the Reynolds number and the solid volume fraction of hybrid nanofluids increased, a rise in the local pressure coefficient was observed. The formation of recirculation zones after and before each rib was visible in the velocity contours. The low-temperature zones created by the recirculation flow were also seen clearly before and after each rib.

The presented simulations were two-dimensional; the generalization to three-dimensional simulations is left for a future study. In addition, the variation of material properties with temperature was neglected. The importance of the temperature-dependence material properties needs to be addressed in the future.

## Figures and Tables

**Figure 1 nanomaterials-12-02720-f001:**
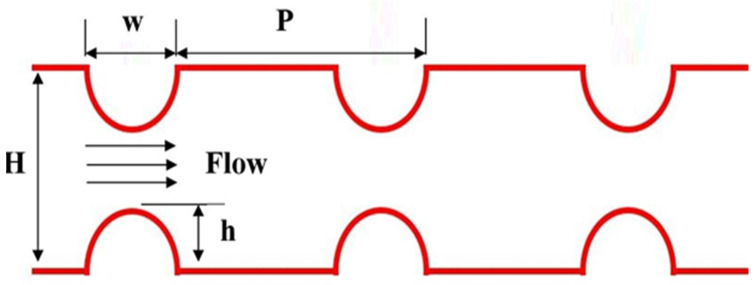
Schematic diagram of a channel with semicircle ribs [23].

**Figure 2 nanomaterials-12-02720-f002:**
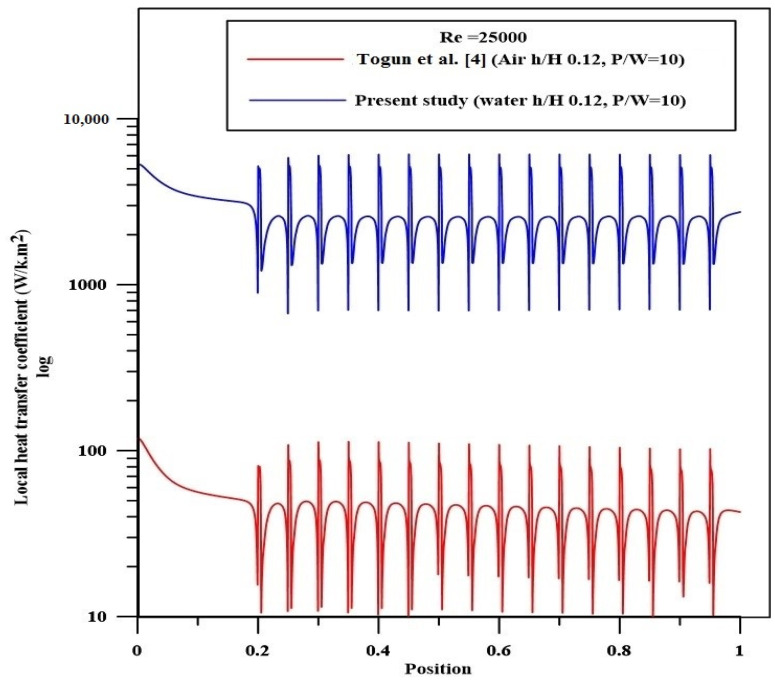
Evaluation of profile of local heat transfer coefficient with Togun et al. [4].

**Figure 3 nanomaterials-12-02720-f003:**
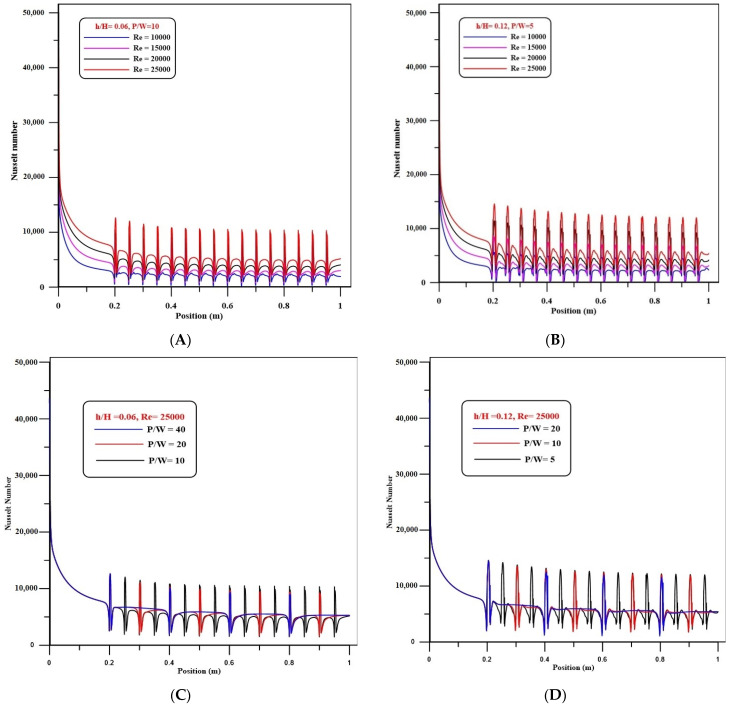
(**A**) Distributions of Nusselt number with different Reynolds numbers of h/H = 0.12 and P/W = 10. (**B**) Distributions of Nusselt number with different Reynolds numbers of h/H = 0.12 and P/W = 5. (**C**) Influence of pitch ratio on Nusselt number for h/H = 0.06 at Re = 25,000. (**D**) Influence of pitch ratio on Nusselt number for h/H = 0.12 at Re = 25,000.

**Figure 4 nanomaterials-12-02720-f004:**
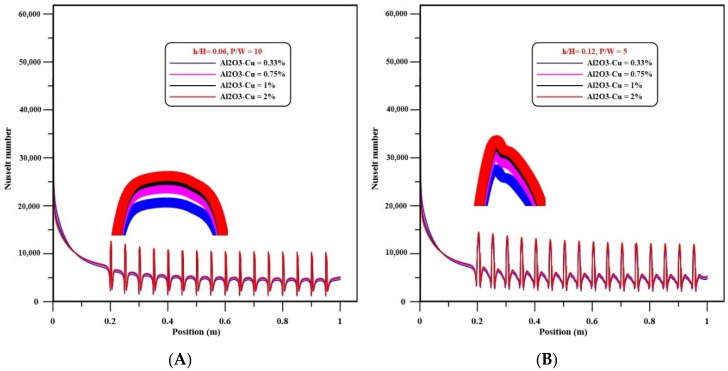
(**A**) Variations of Nusselt number with different hybrid nanofluids of Al_2_O_3_-CuO at h/H = 0.06 and P/W = 10. (**B**) Variations of Nusselt number with different hybrid nanofluids of Al_2_O_3_-CuO at h/H = 0.12 and P/W = 5.

**Figure 5 nanomaterials-12-02720-f005:**
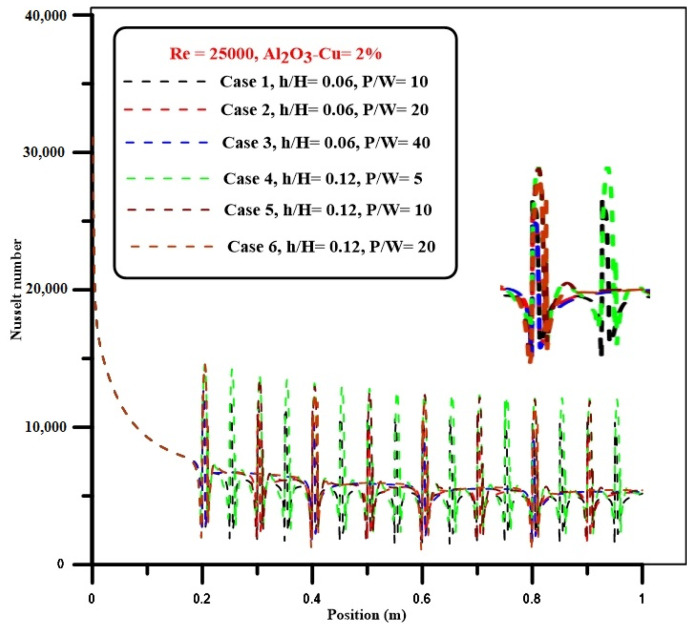
Comparisons between all cases at various step heights and pitch spacing for the ribs at Re = 25,000.

**Figure 6 nanomaterials-12-02720-f006:**
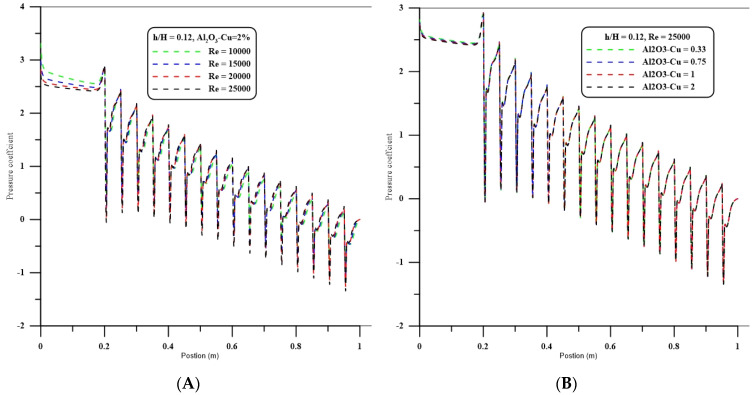
(**A**) Effect of Reynolds number on pressure coefficient for %2 hybrid nanofluids of Al_2_O_3_-CuO at h/H = 0.12 and P/W = 10. (**B**) Effect of volume fraction of hybrid nanofluids of Al_2_O_3_-CuO for Re = 25,000 at h/H = 0.12 and P/W = 10.

**Table 1 nanomaterials-12-02720-t001:** Dimensions of ribs for six cases.

Case	h	h/H	P	P/W
1	2.5	0.06	50	10
2	2.5	0.06	100	20
3	2.5	0.06	200	40
4	5	0.12	50	5
5	5	0.12	100	10
6	5	0.12	200	20

**Table 2 nanomaterials-12-02720-t002:** Constants of SST K-ω turbulence model [44].

$\sigma_{k,1}\;=\;1.176$	$\sigma_{k,2}\;=\;1$	$\sigma_{\omega ,1}\;=\;2$	$\sigma_{\omega ,2}\;=\;1.168$	$\beta_{i,1}\;=\;0.075$
$\beta_{i,2}\;=\;0.0828$	$R_{K}\;=\;6$	κ = 0.41	$\beta_{\infty }^{*}\;=\;0.09$	

**Table 3 nanomaterials-12-02720-t003:** Physical properties of water and nanoparticles [46].

Physical Properties	Water	Cu	Al_2_O_3_
c_p_/J kg^−1^ K^−1^	4179	385	765
ρ/kg m^−3^	997.1	8933	3970
k/W mK^−1^	0.613	400	40
β^−1^_K_-1	21 × 10^−5^	1.67 × 10^−5^	0.85 × 10^−5^
δ/Ωm^−1^	0.05	5.96 × 10^7^	1 × 10^−10^

**Table 4 nanomaterials-12-02720-t004:** Thermophysical properties of hybrid nanofluid [47].

φ%	k_nf_ (W/m K)	$\mu_{\mathrm{nf}}$ (kg/m s)
0.10	0.6199817	0.000972
0.33	0.6309797	0.001098
0.75	0.6490042	0.001386
1.00	0.6570083	0.001602
2.00	0.6849921	0.001935

**Table 5 nanomaterials-12-02720-t005:** Grid independence tests.

Grid	Number of Grid Nodes	h_av_ _(W/m^2^·K)_
1	40,480	1457.51
2	92,400	1559.57
3	165,760	1602.178

## Data Availability

Not applicable.

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
