# Peer review of "Efficient Heat Transfer Augmentation in Channels with Semicircle Ribs and Hybrid Al2O3-Cu/Water Nanofluids"

_nanomaterials, 2022, doi:10.3390/nano12152720_

Round 1
Reviewer 1 Report
This is a very interesting work. Some comments are included below:
· Page 3: Please refer to importance of the assumptions and the errors they possibly lead to.
· Please explain the basic symbols used in equations (1)-(24), some of them are not obvious.
· The results of Fig. 5 and 6 are not clear, please improve analysis.
· Are there any experimental results from your work or in bibliography which is in accordance with the simulation results presented in this study?
Author Response
Thank you for your constructive comments that improved the quality of the paper. Please find enclosed our responses.

Reviewer 2 Report
In this paper, the authors studied the heat transfer enhancement of hybrid "Al2O3-Cu/water" nanofluids flowing in a two-dimensional channel with semicircle ribs. A computer modeling approach using a finite volume approach with k-ω shear stress transport turbulence model was used in these simulations. A higher number of ribs was also shown to result in a larger number of recirculation zones, increasing the thermal performance. The studied topic is very important, and the paper is well organized. However, many theoretical issues are mixed, and the conclusion is intuitive.
· The authors should show more about their numerical algorithm.
· Although it is understandable that the study only considered two dimensional case, the extension to more general 2d or 3d should be of great interest to the readers and should at least be commented on. For example, how would this method scale with the problem size and dimension? Does the architecture require any fundamental change to capture 3d phenomena?
· The applied two dimensional is axial symmetric? Or like two parallel plates with ribs?
· There are some typo in the paper. I suggested the checked the paper again and correct them.
· There are some disordered of the figure format of the manuscript, please check carefully. Enhance the resolution of figures.
· Some of Table is presented as figure which is unacceptable.
· The authors should add the nomenclature section in order to define all variables which is used in the manuscript.
· Please reorganize the conclusion section by deep discussing the results rather than simply using bullet formats. Furthermore, seems some paragraphs have been missed.
Author Response

(The authors gave the same response as above.)

Round 2
Reviewer 2 Report
I am satisfied with the revision.